# A Surprising Diversity of Xyloglucan Endotransglucosylase/Hydrolase in Wheat: New in Sight to the Roles in Drought Tolerance

**DOI:** 10.3390/ijms24129886

**Published:** 2023-06-08

**Authors:** Junjie Han, Yichen Liu, Yiting Shen, Weihua Li

**Affiliations:** College of Agriculture, The Key Laboratory of Oasis Eco-Agriculture, Xinjiang Production and Construction Group, Shihezi University, Shihezi 832003, China; hanjunjie1208@sina.com (J.H.); lyc162710@163.com (Y.L.); 15662712664@163.com (Y.S.)

**Keywords:** wheat, *XTH* gene family, drought, root plasticity, transgenic

## Abstract

Drought has become a major limiting factor for wheat productivity, and its negative impact on crop growth is anticipated to increase with climate deterioration in arid areas. Xyloglucan endoglycosylases/hydrolases (XTHs) are involved in constructing and remodeling cell wall structures and play an essential role in regulating cell wall extensibility and stress responses. However, there are no systematic studies on the wheat *XTH* gene family. In this study, 71 wheat *XTH* genes (*TaXTHs*) were characterized and classified into three subgroups through phylogenetic analysis. Genomic replication promoted the expansion of *TaXTHs*. We found a catalytically active motif and a potential N-linked glycosylation domain in all TaXTHs. Further expression analysis revealed that many *TaXTHs* in the roots and shoots were significantly associated with drought stress. The wheat *TaXTH12.5a* gene was transferred into Arabidopsis to verify a possible role of *TaXTHs* in stress response. The transgenic plants possessed higher seed germination rates and longer roots and exhibited improved tolerance to drought. In conclusion, bioinformatics and gene expression pattern analysis indicated that the *TaXTH* genes played a role in regulating drought response in wheat. The expression of *TaXTH12.5a* enhanced drought tolerance in Arabidopsis and supported the XTH genes’ role in regulating drought stress response in plants.

## 1. Introduction

The rapid increase in the global population has determined rapid growth in the demand for cereal-based food. The development of modern agricultural technology and advances in breeding techniques have significantly increased the efficiency of grain production; however, climate change and water wastage due to industrial production are having a massive effect on agricultural production. Kao and Ganguly [1] forecasted changes in precipitation frequency, intensity, duration, and distribution, which could further exacerbate drought stress levels in regions such as the Middle East and Xinjiang, China, and negatively affect wheat cultivars’ growth and development [2]. Drought is one of the major limiting factors for agricultural production, negatively affecting crops at the metabolic level [3]. For example, drought can reduce seed germination rates, seedling growth, yield, quality, etc. As a multidimensional stress, drought can have short-term (minutes) or long-term (months) molecular, physiological, and morphological effects on all stages of plant growth, leading to reduced plant biomass, respiratory and photosynthetic inhibition, impaired organ development and cellular damage, ultimately leading to death [4].

The reconstruction of plant cell walls plays a fundamental role in controlling cell growth and morphogenesis [5]. Among them, xyloglucan endotransglucosylase/hydrolase (XTH) is one of the critical enzymes in plant cell wall remodeling, both by endocytosis of xyloglucan molecules through xyloglucan endotransglucosylase (XET) activity to achieve xyloglucan chain breakage and molecular grafting. Xyloglucan endohydrolase (XEH) activity specifically hydrolyzes the xyloglucan β-1,4 glycosidic bond [6,7], thus modifying the cellulose-xyloglucan complex structure [8]. The prevalence of XTH suggests that it is essential in plant development [9,10,11]. XET activity and *XTH* gene expression have been experimentally confirmed to be closely related to cell proliferation [12,13,14,15]. The role of some *XTH* genes as cell growth promoters has been demonstrated through loss- or gain-of-function experiments [16,17,18].

The relevance of *XTH* in cell growth and cell wall remodeling suggests that it may have practical applications in stress responses. For example, 33 *AtXTHs* in Arabidopsis showed different expression patterns under abiotic stresses such as touch [19], gravity [20], metals [21], wind [22], drought and salt [23], and light [24]. In addition, similar reports have been made on other plants, such as beans [25], rice [26], poplar [27], and tomato [28]. Overexpression of *CaXTH3* in Arabidopsis and tomato can improve drought and salt tolerance in transgenic plants. The enhanced tolerance of transgenic plants to severe water deficit confirms that *XTH* enhances drought tolerance [28,29]. Elevated XET activity contributed to cell wall loosening in maize roots under low water potential conditions, and abscisic acid (ABA) participates in regulation [30]. Gibberellin (GA) induced high expression of *OsXTH8* in leaf sheaths and young roots, and its expression inhibition resulted in stunted growth of transgenic plants [31]. Functional deficiency of *AtXTH31* resulted in reduced ABA sensitivity and accelerated seed germination in Arabidopsis [32]. The homologous gene of *GmXTH16* in maize is regulated by flooding and ethylene and is associated with aeration tissue development [33].

As one of the most important food crops in the world, wheat provides an important food source for humans, but its production is subject to several environmental factors [34]. Among the mainly abiotic stresses, wheat is susceptible to drought, and wheat photosynthesis and yield are greatly limited under water deficit conditions [35]. Drought can cause downregulation of gene expression related to cell wall metabolism, cellulose synthesis, and cell wall degradation, indicating that cell wall biosynthesis is inhibited [36]. Therefore, it is of great interest to investigate the functions of the *XTH* gene family in wheat to reveal the drought resistance mechanism. However, little is known about wheat *XTHs* except for the five reported *XTHs* [37] involved in cell wall expansion. The availability of wheat genome sequences and whole-genome analysis of wheat *XTHs* by transcriptome sequencing, including phylogeny, chromosome distribution, and structural analysis, were performed for verification of the identified *TaXTHs* by qRT-PCR. The dataset provides a complete list of *TaXTHs* and outlines their dynamic expression patterns. Further functional validation of *TaXTH12.5a* was performed to evaluate its potential role in controlling plant resistance to drought stress. This work provides important insights into the functional role of *XTH* genes in wheat under drought conditions.

## 2. Results

### 2.1. Genome-Wide Identification of the TaXTH Family and Phylogenetic Relationships

In general, *XTH* family members have high conservation in plants, so we used the amino acid sequence of *TaXTH1* (GenBank: AY589585.1) as a bait to retrieve all possible unigenes of *TaXTHs* proteins from the Phytozome v13 database and Wheat Expression Browser (http://www.wheat-expression.com/ (accessed on 13 January 2023)), and proteins containing both XET_C and Glyco_hydro_16 structural domains were detected by SMART, yielding a total of 71 best non-redundant candidate genes, which were named according to their position on the chromosome (Appendix A). These putative proteins encoded by *TaXTHs* demonstrated the conserved structural features of the XTHs.

Phylogenetic analysis of plant XTH proteins based on the genetic history and sequence identity of unknown XTH members is a reliable method to understand their functions. Therefore, a phylogenetic tree was constructed using the full-length XTH protein sequences of 71 TaXTHs and 22 Arabidopsis (AtXTHs) with the structurally characterized bacterial lichenase (1GBG, EC3.2.1.73) as the outgroup (Figure 1). Four taxa (Ancestral group, Group Ⅲ, Group Ⅰ, and Group Ⅱ) were identified based on the topology of the ML tree and the previous classification of the XTH family in Arabidopsis [38,39]. It is generally believed that Group Ⅰ and Group Ⅱ members have similar structures and functions. They tend to be grouped and have the most significant number of members, which is consistent with the results of the present study. Group Ⅲ was subdivided into Group Ⅲ-A and Group Ⅲ-B according to the previous definition [40]. So far, Group Ⅲ-A exclusively exhibits XEH activity [41], and five members (TaXTH9.2b, TaXTH9.3d, TaXTH10.1a, TaXTH12.3b, and TaXTH12.5d) were assigned to this group. In wheat, TaXTH8.1a, TaXTH8.3b, TaXTH8.4b, TaXTH12.1b, TaXTH12.3a, and TaXTH12.3d belonged to the ancestral taxon (Figure 1). The extensive expansion of the *TaXTH* gene family in wheat compared with Arabidopsis may correspond to the larger genome of wheat.

### 2.2. Chromosomal Distribution and Duplication Process of TaXTHs

The *TaXTH* genes were localized on wheat chromosomes, and as shown in Figure 2, 71 *TaXTHs* were unevenly distributed on the chromosome. Chromosome 7D carried the most significant number of genes (9), followed by chromosomes 2D (7) and 7A (7), chromosome 2A had 6, and chromosome 2B had 5. Chromosome 3D had no *XTH* genes, while the remaining chromosomes contained one to four *XTH* genes each.

Tandem duplication events are one of the mechanisms of gene amplification. Gene replication has long been considered the leading force in the evolution and expansion of a gene family [42]. Five or fewer genes in the 100 Kb range are generally believed to be tandem duplicates [43]. In the present study, we found 14 *TaXTHXs* distributed in the 100 Kb, located on chromosomes 2A, 2D, 3B, and 7D (Figure 2), suggesting they may be tandem duplicates. The similarity of putative tandem genes was calculated using Smith-Waterman (https://www.ebi.ac.uk/Tools/jdispatcher (accessed on 13 January 2023)) [44] (Appendix A). The results showed that the similarity between *TaXTH7.3a*, *TaXTH7.4a*, and *TaXTH7.5a* was low (58.2–64.3%), while the similarity of the remaining putative tandem genes was high (88.1–97.5%). Based on the phylogenetic tree, these genes were closely related, and we concluded that these genes were tandem duplicates. The Ka/Ks ratio of duplicated gene pairs varied from 0.064–0.498 (Appendix A). This indicates that the mutations in the homologous *TaXTH* genes are neutral or disadvantageous, and tandem duplication and fragment repeats contribute to the amplification of the *TaXTH* genes. In addition, four tandem duplications were also found in Arabidopsis. For example, *AtXTH1/2*, *AtXTH23/14*, and *AtXTH24/18/19* on chromosome At04 and *AtXTH12/13/25/22* on chromosome At05 are from tandem repeats (Appendix A). Notably, some genes with covariance with these tandem genes have conserved tandem repeat patterns, suggesting that these tandem duplication sequences appeared before the divergence of Arabidopsis and wheat.

### 2.3. Gene Structure of TaXTHs and Structure-Based Protein Conservative Sequence Analysis

Previous studies in Arabidopsis have shown that the explicit substructure of *XTH* is relatively conservative [39]. To better characterize the structural diversity of *TaXTH* genes and the conserved protein structure, we obtained the exon-intron structure of each member through GSDS. We determined the motif diversity of TaXTH proteins by MEME. As shown in Figure 3, fifteen (15) types of motifs from wheat were identified. Each TaXTH protein has a Glyco_hydro_16 and a XET_C structural domain. The Glyco_hydro_16 spans 13-10-12-4-3-11-1-7-6-2 motifs, although some proteins lack one or more. For example, TaXTH7.1d and TaXTH12.1d are short of motifs 13, 10 and 12 and are less than 250 amino acids long. Almost all TaXTHXs share motifs 4, 3, 11, 1, 7, 6, 2, 5 and 9. Overall, the motifs exhibit a similar distribution in the TaXTH protein sequence. In addition, all members of the *TaXTHs* family contain two to four exons (Figure 3C). Several of these genes exhibit different structures. For example, *TaXTH6.2a*, *7.1a*, *7.3a*, *7.4a*, *8.1a*, *8.4b*, *11.3a*, and *12.3a* lack 5′ and 3′ untranslated regions (UTRs), while *TaXTH9.3d*, *12.1b* and *12.4a* have extremely long 5′ UTRs.

All TaXTH proteins have the catalytically active conserved motif DExDxE (Appendix A), so it is reported here that TaXTH may catalyze the breakage/reconnection of xyloglucan molecules or catalyze xyloglucan hydrolysis. In addition, this conserved motif can be extended upstream and downstream, showing higher conservation (Appendix A). A potential N-linked glycosylation site sharing N(T/S)V(L/S/R/E/A/Q/I)T(S/R/D/K/P/W)G was located near this motif in 71 TaXTH proteins. This site binds N-glycans and is associated with protein stability, as reported in a *Fragaria vesca* study [45].

To gain a more detailed understanding of the structural characteristics of the TaXTH protein, ESPript (https://espript.ibcp.fr/ESPript/ESPript/index.php (accessed on 14 January 2023)) was used to compare the TaXTH proteins with the xyloglucan endoglycosyltransferase PttXET16-34 (PDB id: 1UN1) [46]. Although all TaXTH proteins identified in this study have two conserved structural domains (Glyco_hydro and XET_C), some of the XTHs lacked one or several α-helices/β-jellyroll compared with PttXET16-34. The comparison results showed that multiple motifs were covered by different α-helices or β-jellyroll, but there is no uniform correspondence between the two. For more detailed information, see (Appendix A). Overall, the motif patterns in the different TaXTH proteins show only minor differences and have similar gene structure patterns.

### 2.4. Expression Pattern of the XTH Genes in the Wheat Germinated Seeds

In order to examine the expression pattern of the *TaXTHs* in the germinating seeds subjected to drought stress, we detected the expression of 71 *TaXTH* genes. As shown in Figure 4 and Appendix A, the members of the *TaXTHs* had specific expression profiles at different stress times. For example, *TaXTH8.4b* and *TaXTH10.1d* were expressed mainly after 24 h of drought stress; *TaXTH7.3b*, *TaXTH7.5a*, *TaXTH9.2b*, *TaXTH11.4b*, and *TaXTH12.5d* were highly expressed at 48 h of drought stress; while *TaXTH7.1b*, *TaXTH7.1d*, *TaXTH8.1b*, *TaXTH9.1b*, *TaXTH 12.6a* and *TaXTH12.7a* were up-regulated at 72 h. The highest number of members significantly expressed under all drought treatments was recorded at 48 h (37), followed by 72 h (20). In addition, the *TaXTHs* were expressed at lower levels under normal conditions, suggesting that members of the *TaXTHs* family respond differently to different durations of drought.

### 2.5. Expression Pattern of TaXTHs in Wheat Roots and Shoots under Drought Stress

To further assess whether the expression profile of the *TaXTHs* changed under drought stress, we performed qRT-PCR on the expression of 36 of these *TaXTH* genes in roots and shoots. In potted plants after 24 h of drought stress, the expression of 21 *TaXTHs* in the roots was significantly higher than that of the control by 2.02, 3.79, 1.95, 1.91, 1.79, 3.55, 2.98, 2.19, 1.5, 2.13, 4.75, 1.85, 4.12, 2.20, 1.86, 1.49, 1.62, 2.71, 1.61, 2.41, and 1.73-fold. In the shoots, 29 *TaXTHs* were up-regulated at the beginning of drought stress, while *TaXTH7.3a* and *TaXTH12.1d* were up-regulated after 48 h. *TaXTH9.2b/11.1b/11.4b* were continuously down-regulated. The expression trends of the 19 *TaXTHs* were identical in the roots and shoots, with *TaXTH12.5a* possessing the largest fold change after 48 h of drought stress, 8.89 and 11.21 times higher than the control, respectively. The expression patterns of the remaining genes in roots and shoots differed depending on the time of drought (Figure 5 and Appendix A).

### 2.6. Stable Transgenic Arabidopsis with Overexpression of TaXTH12.5a

Identification of promoter cis-regulatory elements of the *TaXTH* genes showed that a 6-bp cis-acting element involved in drought-inducibility (CAACTG) was enriched in most *TaXTHs* (Appendix A). All *TaXTH* genes contain multiple response elements. The main effects of drought include photosynthesis, respiration, and hormones, which severely affect natural plant growth and assimilation/water uptake [47]. These results suggest that the *TaXTH* gene family may play an essential role in plant tolerance to drought stress.

To further investigate how the *TaXTH* genes affect plant morphology in plant development and maintain physiology under drought stress, *TaXTH12.5a* was transferred to Arabidopsis. More than 40 independent transgenic lines were obtained, and PCR tests were performed on 8. Subsequently, the copy number changes of the transgenes in these events (T0 generation) were inferred using the Q3D PCR. Three events (code: OE-40-3, OE-40-7, and OE-40-14) contained one copy number of the transgene, and one event (Code: OE-40-16) had two copies (Figure 6A). However, the other four events had low copy numbers (<0.5), predicting that they might be chimeras (transgenic plants consist of two or more different cell lines). T3 generation pure lines of the above four transgenic events were obtained and screened for resistance using thaumatin. qRT-PCR detected the transcript abundance of the four T3 generation lines, and lines OE-40-7 and OE-40-16 increased 23 and 16 times compared with the non-transgenic plants, respectively. At the same time, OE-40-3 and OE-40-14 showed a more significant increase (Figure 6B).

### 2.7. Transgenic Arabidopsis Exhibits Drought Tolerance during the Seedling Stage

In the present study, we compared the tolerance of transgenic and control lines under drought stress and counted the germination rate and root length. As shown in Figure 7, after 5 days of drought, the transgenic lines possessed longer roots and hypocotyls (3.1 to 3.8 cm) compared with the control (2.8 cm) (Figure 7A). Meanwhile, four T3 transgenic lines with different expressions showed some drought tolerance, increasing the germination rate from 44% to 68%. Among them, three lines (OE-40-3, OE-40-7, and OE-40-14) showed a significant increase compared to the control germination rate of only 32% (Figure 7B). Root and hypocotyl lengths were significantly longer in both lines (OE-40-7 and OE-40-16) than in the control (Figure 7B). These results suggest that Arabidopsis overexpressing *TaXTH12.5a* possesses enhanced tolerance to drought stress as reflected by higher germination rates and longer roots and hypocotyls compared with the control.

### 2.8. Transgenic Arabidopsis Exhibits Drought Tolerance during the Vegetative Stage

Similarly, the ability of transgenic plants to withstand drought during the vegetative stage was explored. Seedlings at two weeks of germination were subjected to 5-day drought treatment (Figure 8A). We found that the transgenic plants possessed a higher green leaf rate under drought stress, predicting higher drought tolerance of the transgenic plants. Moreover, the primary roots of the transgenic plants were longer than those of the control (Figure 8B), which was more evident in lines OE-40-7 and OE-40-16. In addition, the transgenic plants exhibited a higher number of lateral roots (Figure 8C). In addition, we observed that the two transgenic lines (OE-40-3 and OE-40-16) demonstrated many tertiary root tips as a response to drought compared with the wild type (Figure 8D). Thus, ectopic expression of *TaXTH12.5a* in Arabidopsis can alter the root architecture of different types of Arabidopsis in soil, thereby enhancing resistance to drought stress.

## 3. Discussion

According to reports, the *XTH* gene family has been identified in many seed plants, including Arabidopsis [39], cabbage [48], cotton [49], rice [26], poplar [27], and tomato [50]. Disappointingly, there are only sporadic reports about *XTH* genes in wheat. In this study, we performed genome-wide identification and characterization of the wheat *XTH* gene family and compared it with the Arabidopsis *XTH* genes. The expression pattern results suggest that *TaXTHs* may be essential in enhancing plant tolerance to drought stress. Transgenic Arabidopsis overexpressing *TaXTH12.5a* showed a significant increase in drought tolerance, accompanied by an increase in primary root length, secondary root number, and tertiary root number.

### 3.1. Characterization of the TaXTH Gene Family

Although the role of the *XTH* gene family in plant cell wall elongation has been demonstrated in numerous studies, knowledge of the size and mechanism of action of the *XTH* gene family in wheat still needs to be improved. In this study, we comprehensively analyzed the wheat (*Triticum aestivum*) genome to explore the composition and functional characteristics of the wheat *XTH* gene family. We classified the *TaXTH* genes through phylogenetic analysis: Group I, Group Ⅱ, and Group Ⅲ. Group I and Group Ⅱ formed the largest cluster, referred to as Group I/Ⅱ, probably due to their evolutionary and functional similarities (Figure 1). However, we noted functional differences between Group Ⅲ and Group I/Ⅱ. In particular, Group Ⅲ-A exhibited XEH activity, whereas Group Ⅲ-B showed XET activity [41]; this suggests that Groups Ⅲ-A and Ⅲ-B may perform different functions in different aspects of cell wall elongation.

Interestingly, our study found that the expansion of the gene family of wheat *XTHs* was partly due to tandem duplication events. This gene family expansion may be related to wheat’s environmental adaptation needs. The cell wall plays an essential function in plant growth and development, and *XTHs*, as cell wall relaxases, play a vital role in the remodeling and extension of the cell wall. Thus, the expansion of the *XTH* gene family in wheat may reflect the evolution of the genome in response to the need for adaptation to the environment and adaptive growth. TD events may have led to the generation of new functional derivations and regulatory mechanisms, further enriching the diversity and function of the *TaXTH* genes.

Regarding functional localization, we found that all TaXTH proteins localize to the cell wall, which is consistent with the function of XTH proteins involved in cell wall remodeling. However, we also found some XTH proteins localized in the cytoplasm (Appendix A). This observation is similar to previous findings in barley [51]. Such cytoplasmically localized XTH proteins may have specific functions and regulatory mechanisms that deserve further study and exploration.

The conserved motifs of XTH proteins have been the subject of much attention in past studies. Our results support this, which found that almost all identified TaXTHs have several specific conserved motifs, including hydrophobic amino acid regions that may act as signal peptides and a highly conserved DEIDFEFLG structural domain, catalytic sites for XET and XEH activity. Notably, all TaXTHs in this study exhibited the highly conserved DExDxE motif near the N-linked glycosylation site (Appendix A). This finding was also supported when compared with the catalytic structure of Arabidopsis XTHs [39]. In addition, we observed some substitutions among the amino acid residues at the catalytic site, such as the substitution of the fifth phenylalanine (F) residue by leucine (L), isoleucine (I) or methionine (M) and the substitution of the seventh phenylalanine (F) and eighth leucine (L) residues by leucine (L) and methionine (M) residues, respectively (Appendix A). As these substituted amino acid residues remain non-polar and uncharged, Fu et al. [51] speculated that these substitutions have little effect on the cleavage of the xyloglucan-glycan chain.

Another striking finding is that most (13 out of 15) motifs are conserved in wheat. Although it is unclear what effect these motifs have on the function of TaXTHs, we can speculate that the presence of different motifs may mean that TaXTHs have distinct biochemical and biological processes in response to abiotic stress. This finding provides new clues for further studies on the role of TaXTHs in adversity response and plant development.

In summary, although our understanding of the mechanism of action and function of the *TaXTH* gene family in wheat is limited, through this study, we have identified several conserved motifs and features of the catalytic site and have proposed the hypothesis that TaXTHs may have different biochemical and biological functions in response to abiotic stress. Further studies will help reveal the detailed processes and regulatory mechanisms of TaXTHs and provide breakthrough points for crop improvement and adversity adaptation in agriculture.

### 3.2. The Expression Patterns of TaXTHs Were Regulated by Drought

Drought may be a primary stressor that hampers plant growth and limits productivity [3]. Plants employ various effective strategies to mitigate the negative impacts of excessive dehydration. Currently, most studies on water stress signaling have focused on salt stress due to the similar effects of salt and drought stress, with overlapping signaling pathways [52]. A recent study showed that when seedlings of *Salicornia europaea* were exposed to different concentrations of NaCl, 27 and 15 *SeXTH* genes were highly expressed in shoots and roots after 24 h at 200 mM NaCl, respectively [53]. Meanwhile, *MtXTH3* showed high expression in the shoots and roots of *Medicago truncatula* under 150 mM NaCl [54]. However, identifying drought-tolerance genes has proven challenging thus far. Therefore, studying the expression patterns and functions of the *TaXTH* gene family can provide valuable insights into how wheat adapts to drought stress.

In the present study, we observed notable variations in the expression patterns of *TaXTHs* in response to different durations of drought exposure and showed tissue specificity. For example, in seeds germinating for two days, the expression of *TaXTH* genes remained minimal under non-drought conditions. However, upon subjecting the seeds to 24 h of drought stress, individual genes (e.g., *TaXTH8.4b* and *TaXTH10.1d*) exhibited significantly elevated expression levels. Furthermore, as drought exposure was prolonged, more *TaXTH* genes were demonstrated to be strongly expressed (Figure 4). These findings indicate that the expression of most *TaXTH* genes is induced by the duration of drought, implying their potential involvement in conferring drought resistance. Moreover, drought duration induces most *TaXTH* gene expression, suggesting that several *TaXTH* genes may be involved in drought resistance and that functional redundancy based on expression patterns may not be present.

To gain deeper insights into the potential roles of *TaXTHs* under drought stress, we analyzed the expression patterns of 36 *TaXTH* genes in wheat roots and shoots. Previous studies have highlighted the crucial role of *CaXTH3* gene overexpression in enhancing drought resistance in species such as *S. lycopersicum* [23] and Arabidopsis [29] during drought in seedlings. In the present study, rigorous expression analysis also showed that 29 and 25 *TaXTH* genes were up-regulated and expressed in roots and shoots at 72 h of drought, respectively. Therefore, it is reasonable to assume that multiple *TaXTH* genes in wheat may have functions in response to drought stress. Furthermore, in shoots, we observed the down-regulation of *TaXTH9.2b*, -*11.1b*, and *-11.4b* at the onset of drought stress, which parallels the expression patterns of several *XTH* genes (*AtXTH6*,-*9*,-*15*, and -*16*) in Arabidopsis when subjected to drought stress [55]. Additionally, a separate study reported the down-regulation of *XTH* genes in the elongation zone of soybean seedlings experiencing low water potential [56]. The differential expression of *XTH* genes in roots and shoots depends on the duration of drought stress, as indicated by Tenhaken [57], who reported distinct expression patterns of *XTH* genes in the shoots and roots of Arabidopsis under 24 h drought stress. Therefore, *XTH* gene-specific expression may facilitate the precise modulation of well-defined topological regions within the plant cell wall, thereby enhancing plant tolerance to drought. Supporting this perspective, Zhu et al. [58] investigated the abundance of various cell wall proteins, including XTHs, in maize seedling roots under drought treatment. Specifically, two XTHs decreased in the first 3 mm of the root tip, one XTH decreased throughout the root, and one XTH showed a slight increase in the 3–7 mm root tip region. These findings strongly indicate that *TaXTH* genes hold significant potential as valuable tools for enhancing drought tolerance in plants.

### 3.3. The Biological Function of TaXTH12.5a in Arabidopsis under Drought Stress

Recent publications have indicated that heightened XET activity can stimulate the cell division rate or induce the loosening of cell wall polymers, facilitating root growth and development. This mechanism is believed to be crucial for enhancing drought tolerance in plants, as it enables stressed plants to allocate additional resources to explore the soil for residual water [59]. Wu et al. [60] reported a similar finding in maize, highlighting an increase in XET activity in the root tip region under drought stress, suggesting the involvement of *XTH* in enhancing cell wall extensibility in that specific region. Furthermore, studies conducted on wheat seedlings subjected to 20% PEG4000-induced stress revealed that root elongation was initially suppressed within 24 h and significantly delayed after 48 h, underscoring the criticality of the initial phase of drought stress on wheat root growth [61]. Collectively, these findings provide evidence for the direct involvement of *XTH* in plants’ response to drought stress. In this paper, we aimed to investigate the seedling growth phenotype of Arabidopsis plants harboring the *TaXTH12.5a* gene under drought stress. *TaXTH12.5a* belongs to Group Ⅱ and is predicted to possess XET activity. Quantitative RT-PCR analysis demonstrated marked upregulation of the *TaXTH12.5a* gene in roots (8.89-fold) and shoots (11.21-fold) after 48 h of drought treatment, exhibiting a specific expression pattern. Interestingly, the enhanced expression of *AtXTH15*,*16*, which share high homology with *TaXTH12.5a*, in roots and shoots corresponded with accelerated plant growth [62]. Consequently, we chose the *TaXTH12.5a* gene for heterologous overexpression in Arabidopsis.

Our investigation revealed that transgenic Arabidopsis exhibited elongated primary roots and embryonic axes and increased adventitious and lateral root formation, indicating enhanced tolerance to drought stress during the seedling and vegetative stages. Previous studies have associated root length, lateral root density, and root hair development with augmented drought tolerance in wheat [63]. Thus, our findings suggest that *XTH*-mediated regulation of cell wall properties plays a pivotal role in plant adaptation to drought stress. Moreover, the *TaXTH12.5a* gene holds potential as a candidate gene for breeding drought-tolerant crop varieties through transgenic breeding. However, further comprehensive investigations are warranted to elucidate the detailed mechanisms underlying its contribution to drought tolerance.

## 4. Materials and Methods

### 4.1. Identification, Chromosomal Location, and Structural Organization of TaXTHs in Wheat

The Phytozome v13 database (https://phytozome-next.jgi.doe.gov (accessed on 13 January 2023)) was retrieved using the BLASTP algorithm, and candidate genes were aligned with IWGSC RefSeq v2.1 (https://urgi.versailles.inrae.fr/blast_iwgsc/?dbgroup=wheat_iwgsc_refseq_v2.1_chromosomes&program=blast (accessed on 13 January 2023)) to obtain the chromosome positions of each *TaXTH*. The conserved structural domains of the candidate *TaXTHs* were identified using SMART (http://smart.embl-heidelberg.de/ (accessed on 13 January 2023)), and those containing both XET_C and Glyco_hydro_16 structural domains were preserved. They were named according to their position on the chromosome. Exon-intron structure visualization was performed based on exon and genomic sequence information and through the Gene Structure Display Server (GSDS) tool. Chromosome location was displayed by TBtools (https://github.com/CJ-Chen/TBtools (accessed on 13 January 2023)).

### 4.2. TaXTH Protein Sequence Alignment and Phylogenetic Analysis

Altogether 71 protein sequences encoding *TaXTHs* were identified for phylogenetic analysis, while bacterial lichenase 1GBG with structural features of GH16 was utilized for the outgroup. Multiple sequence alignment and removal of signal peptide sequences were performed using the default parameters of the Clustal W2 (http://www.ebi.ac.uk/tools/clustalw2 (accessed on 13 January 2023)) tool [64]. The Maximum Likelihood (ML) algorithm generated the phylogenetic tree with the default parameters of PhyML 3.0. The percentage of trees in which related taxa clustered together was determined from 1000 bootstrap replicates. Evolutionary distances were calculated using the Poisson correction method. The tree was visualized using Itol v6 [65].

### 4.3. Plant Growth Conditions and Treatments

Spring wheat cultivar ‘Xinchun 11’ was used for gene expression pattern analysis. Sterilized and cleaned seeds of uniform size were grown in Petri dishes under dark conditions for two days. To study the effect of drought stress on the gene expression of *TaXTHs* during seed germination, a portion of germinating seeds was selected and exposed to −0.5 MPa of D-sorbitol for 0, 24, 48, and 72 h. The concentration of D- sorbitol was determined by referring to the method of Lü et al. [66]. Other seeds were transferred to containers of sterilized sand and soaked in ½ Hoagland nutrient solution at a temperature of 10 °C (9 h day)/7 °C (15 h night), relative humidity of 60%, and a photon flux of 500 μmol m^−2^ s^−1^. After two weeks, the conditions were switched to 28 °C (continuously unchanged) with a relative humidity of 55%, and these conditions were maintained until the sampling period. At the elongation stage, after 0, 24, 48, and 72 h of drought using the above method, shoots, and roots were collected separately and quickly frozen in liquid nitrogen and stored at −80 °C for further analysis.

The genetically modified and wild-type Arabidopsis seeds were spread evenly in a box with three filter paper layers, and then a −0.5 MPa of D-sorbitol solution was added. The box was placed in an artificial climate box for growth. Approximately 10 mL of D-sorbitol solution was added to the filter paper daily and the excess solution was absorbed for five consecutive days. The germination standard is to break through the seed coat by about 1 mm at the embryonic root, and the germination rate is calculated. After disinfection and cleaning, the Arabidopsis seeds from various transgenic lines were uniformly sown on the MS medium. Following two weeks of cultivation, seedlings with consistent growth were carefully selected and transplanted into containers filled with sterilized sand. Subsequently, they were subjected to continuous watering with a −0.5 MPa of D-sorbitol solution for five days. The primary root length, number of lateral roots, and number of tertiary root tips were quantified for analysis. All samples were collected in biological triplicate.

### 4.4. Structure-Based Sequence Alignment, Construction of Conserved Motifs, and Promoter Analysis

PttXET16-34 (ID:1UN1) [46] was downloaded from the PDB (https://www.rcsb.org (accessed on 13 January 2023)) database to determine the secondary structure, and the online tool ESPript 3.0 (https://espript.ibcp.fr/ ESPript/cgi-bin/ESPript.cgi (accessed on 13 January 2023)) [67] was used to predict the protein structural elements, aiming to identify common structural elements in TaXTHs. In addition, MEME (https://meme-suite.org/meme/tools/meme (accessed on 13 January 2023)) [68] was employed to characterize the conserved motifs of the TaXTH proteins, and they were visualized by TBtools. Gene sequences 2000 bp upstream of the translation start site (ATG) were extracted from the wheat genome and analyzed for the presence and abundance of cis-elements in promoter sequences with the help of PlantCARE (http://bioinformatics.psb.ugent.be/webtools/plantcare/html/ (accessed on 13 January 2023)) [69].

### 4.5. RNA Isolation, cDNA Transcription, and Quantitative Real-Time PCR for the TaXTH Genes

In accordance with the manufacturer’s instructions, RNA was extracted from 100–200 mg frozen tissue using a TransZol Up Plus RNA Kit (Lot# Q41020). The quality and quantity of the extracted RNA were assessed using a Nanodrop 8000 (Thermo Fisher Scientific Inc., Logan, UT, USA) and an Agilent Bioanalyzer 2100 (Agilent Technologies Inc., Santa Clara, CA, USA), respectively. EasyScript One-Step gDNA Removal and cDNA Synthesis Super Mix (Lot# P20708) were used for reverse transcription. The *Actin* gene (GenBank accession number: KC775782.1) was used as a reference gene. The control group consisted of plants of the same age without drought treatment. According to the manufacturer’s protocol, three independent biological replicates were performed using the PerfectStartTM Green qPCR SuperMix (TransGen Biotech, Beijing, China). qRT-PCR was performed using the ABI QuantStudio™ 6 system (ABI). The relative gene expression levels were normalized using the 2^−ΔΔCT^ method [70]. The primers used for quantitative PCR are listed in Appendix A.

### 4.6. Construction of the pCAMBIA1301-TaXTH12.5a Vector, Agrobacterium-Mediated Arabidopsis Transformation and Segregation of Posterity

Specific primers were designed to isolate the *TaXTH12.5a* full-length CDS from the wheat using 5′-ACGGGGGACTCTTGA*CCATGG*ATGGCGGTGTCGGTGCTG-3′ (italicized as *NcoI*) and 5′-GTCACCTGTAATTCA*CACGTG*CTACATGGAGCACTCGGCGG-3′ (italicized as *PmlI*). The PCR product (870 bp) was inserted into the pMD18-T vector, and the positive plasmid underwent whole sequencing. The *TaXTH12.5a* gene was cloned into the pAHC25 vector containing the Ubi promoter upstream of its start codon (ATG). Then, the entire gene box was transferred into the pCAMBIA1301 vector. The validated pCAMBIA1301-*TaXTH12.5a* vector was transferred into Arabidopsis by the Agrobacterium-mediated method. To determine the segregation of transgenic plants and target genes, at least 40 plants were selected from the T0 generation for GUS staining and resistance (150 μg/mL hygromycin, Vetec, Sigma, Shanghai, China) screening. Similar analyses were performed on T1 and T2 generation plants to identify pure lines for subsequent studies.

### 4.7. DNA Extraction and PCR Identification of Transgenic Plants

According to the manufacturer’s instructions, DNA from transgenic plants was extracted using an EasyPure Plant Genomic DNA Kit (Lot# R10110), and its purity and quality were tested using Nanodrop 8000. The DNA concentration in the PCR reactions was quantified by a Qubit dsDNA HS Assay Kit (Lot# Q32854) on a QubitTM Flex Fluorometer (Thermo Fisher Scientific, USA). We designed the detection primers based on the *GUS* and *TaXTH12.5a* sequence (Appendix A). The PCR reaction conditions were set as follows: hot start at 94 °C for 4 min, denaturation at 95 °C for 30 s, annealing at 55 °C for 30 s, extension at 72 °C for 1 min, followed by a final extension at 72 °C for 10 min, 33 cycles. Agarose gel electrophoresis assays showing *GUS* and target genes of 870 bp and 940 bp, were considered transgenic events. Four transgenic plants were used for further phenotypic analysis.

### 4.8. QuantStudio 3D Digital PCR Analysis for the Arabidopsis TaXTH12.5a Transgenic Copy Number

The QuantStudio™ 3D Digital PCR Instrument was used to perform imaging and preliminary analysis of a QuantStudio™ 3D Digital PCR 20K Chip v2 (Cat#: A26317). The chip had to be loaded with fluorescent-labelled quantitative PCR reagents (TaqMan™ probe-based assays) and thermal cycled using a Dual Flat Block GeneAmp™ PCR System 9700 (Cat#: 4339386). QuantStudio™ 3D AnalysisSuite™ Software performed the subsequent analysis and post-processing of the absolute quantification results from the raw imaging data. The dPCR reaction volume was 34.8 µL and contained 17.4 µL Master mix, 1.7 µL TaqMan^TM^ Assays primers/probe mix, 3.5 µL diluted genomic DNA (10 ng/µL), and 12.2 µL nuclease-free water. It was mixed well by being gently pipetted up and down and then 14.5 µL of the reaction product was loaded onto each chip. The imaging data stored on the Connect cloud-based platform was analyzed with AnalysisSuite™ Software. The designed probe can only be used to amplify the transgene *TaXTH12.5a* and not the homologous gene of Arabidopsis. The copy number of *TaXTH12.5a* in the same PCR reaction was calculated as follows: (copies/µL of the *TaXTH12.5a* transgene)/(copies/µL of the lectin gene *Ta. LOC123165130*). The Arabidopsis transgenic plants contained one and two insert copies when the ratio value was equal to 0.5 and 1, respectively. When the ratio was less than 0.5, this means that chimeric transgenic plants were found. The primers and probe sequences are shown in Appendix A.

### 4.9. Data Analysis

SPSS V20 (SPSS, Inc., Chicago, IL, USA) was used for all analyses. Gene expression differences were analyzed using an ANOVA with Fisher’s LSD tests. Differences between means were detected using Duncan’s multiple range tests at a significance level of *p* < 0.05.

## 5. Conclusions

Our findings demonstrate that the wheat genome contains a record-breaking XTH gene family with at least 71 genes, surpassing other known species. Through phylogenetic analysis, chromosomal localization, and gene/protein structure examination, we have comprehensively understood the *TaXTH* gene family in wheat. Tandem duplication events attribute to the expansion of the *TaXTH* genes. The conservation of amino acid sequences and motifs further elucidates their functional roles. Moreover, the duration of drought treatment influences the expression profiles of *TaXTHs* in wheat, exhibiting distinct patterns in roots and shoots. Notably, the heterologous expression of *TaXTH12.5a* in Arabidopsis implies that increased XTH activity enhances seed germination and contributes to drought tolerance by developing longer and more abundant secondary roots. Further extensive experimentation is required to elucidate the precise functions of *XTHs* in wheat.

## Figures and Tables

**Figure 1 ijms-24-09886-f001:**
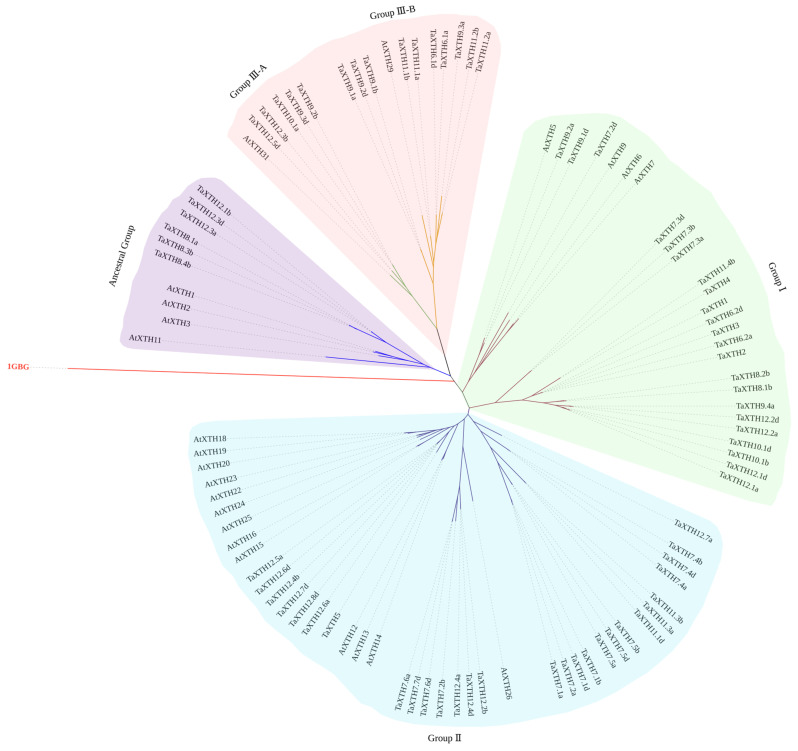
Phylogenetic analysis of full-length XTH proteins in wheat and Arabidopsis. The tree was constructed by maximum likelihood (ML) by PhyML 3.0 and bootstrap values based on 1000 replications. The outgroup is highlighted in red, and the branches with different colors correspond to the four groups.

**Figure 2 ijms-24-09886-f002:**
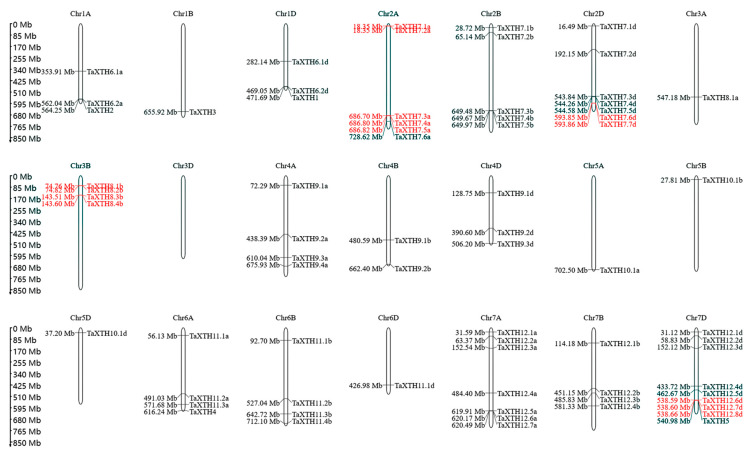
Distribution of *XTH* genes on *T. aestivum* chromosomes. The *TaXTH* genes in red are tandem duplication genes. The number of chromosomes is indicated at the top of each chromosome. The scale on the left is in megabases (Mb).

**Figure 3 ijms-24-09886-f003:**
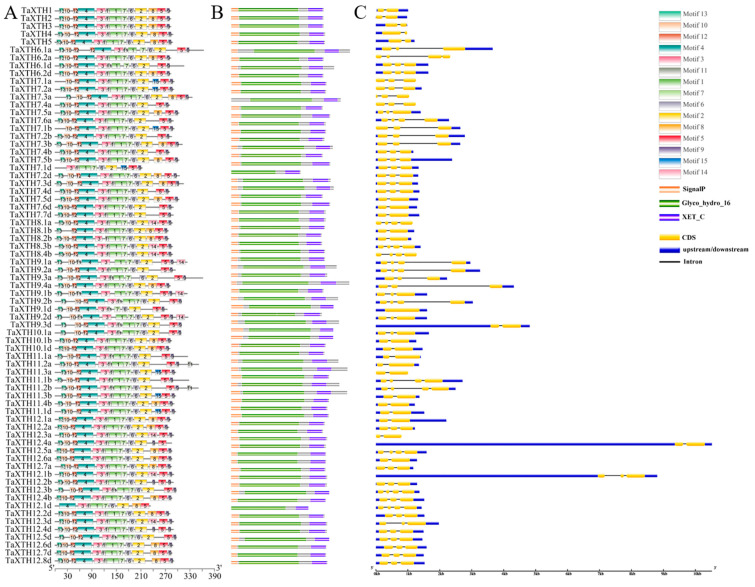
Characterizations of *XTH* gene family members in wheat. (**A**), conserved motif positions. Identification of the motif composition of TaXTH proteins using MEME. Different colored boxes represent different motifs and their positions in the protein sequences. (**B**), domain location. (**C**), exon-intron gene structure features.

**Figure 4 ijms-24-09886-f004:**
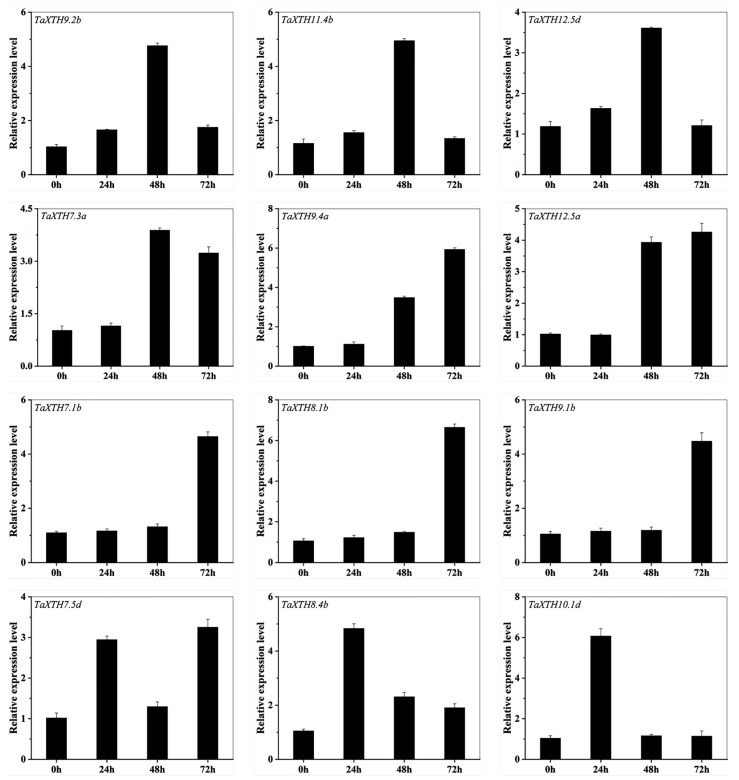
The expression of the *TaXTH* gene in the spring wheat line ‘Xinchun11’ was analyzed using real-time quantitative PCR. The relative expression level of *TaXTH* in wheat was measured after subjecting it to different durations (h) of drought stress with −0.5 MPa D-sorbitol. The 2-day-old germinated wheat seedlings were immersed in −0.5 MPa D-sorbitol and sampled at 0, 24, 48, and 72 h after treatment to assess the expression of *TaXTH*. Genes in the horizontal direction exhibit similar expression patterns. For additional details, see Appendix A.

**Figure 5 ijms-24-09886-f005:**
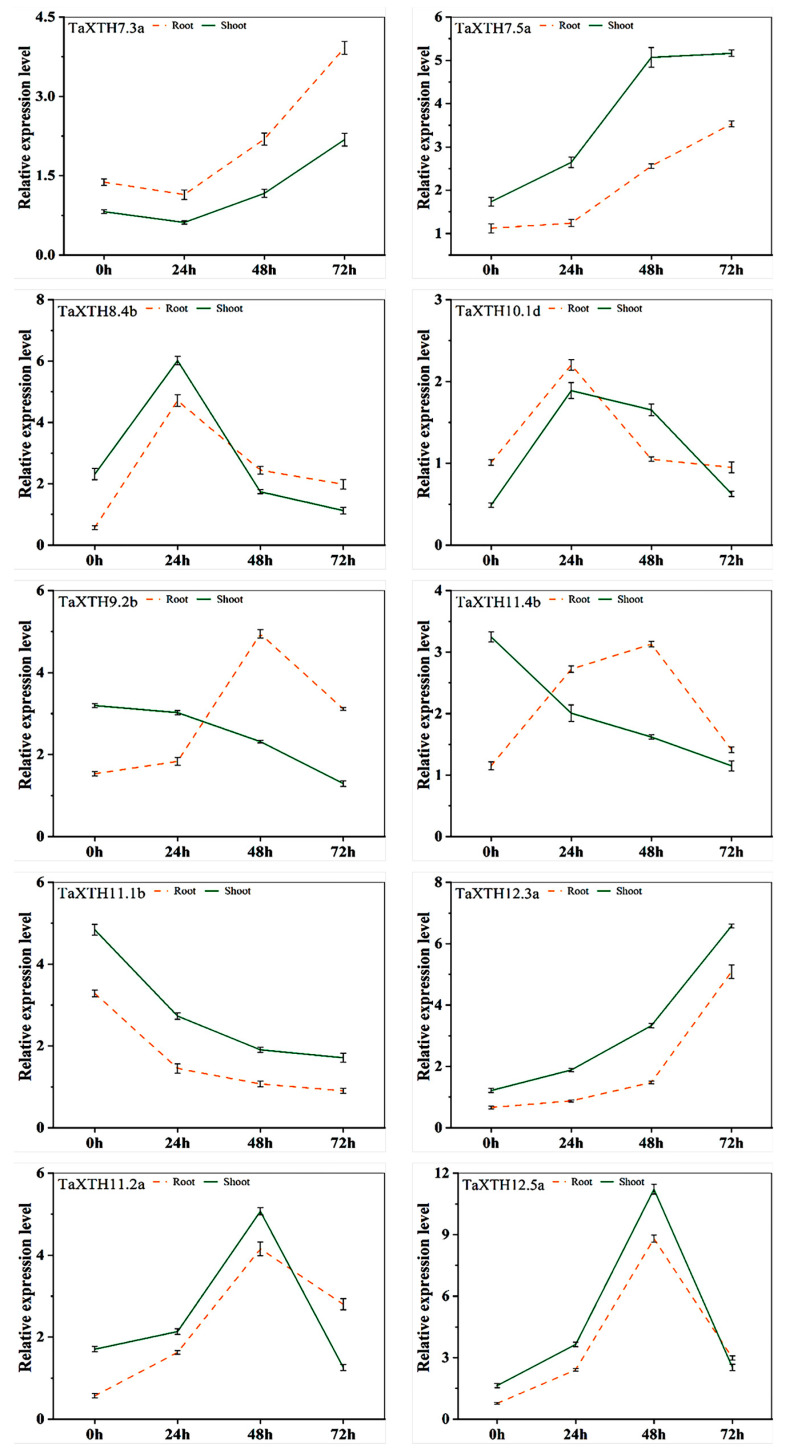
qRT-PCR analysis of *TaXTH* genes expressed under drought stress. Samples for expression profiling were collected from wheat at 0, 24, 48, and 72 h post drought stress. Expression profiles were detected by qRT–PCR and normalized to Actin. Note that the relative expression of *TaXTHs* was on a different scale. Orange dashed lines indicate roots, and green solid lines represent shoots. The name of the gene is shown in the upper left corner of each line graph. Results were analyzed in three biological replicates. Only partial gene expression patterns that were either ‘identical’ or ‘opposite’ were displayed. For additional details, see Appendix A.

**Figure 6 ijms-24-09886-f006:**
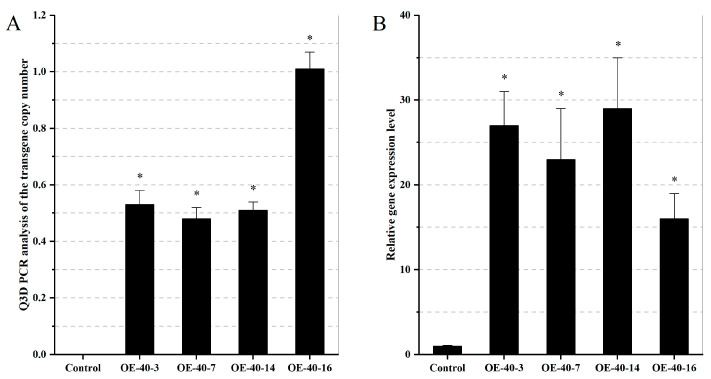
Calculation of copy number and relative expression level of *TaXTH12.5a* in four transgenic events. (**A**) The copy number ratio of *TaXTH12.5a* and lectin gene (*Ta. LOC123165130*) in Arabidopsis T0 transgenic plants was determined using digital PCR. When the ratio is equal to 0.5, transgenic Arabidopsis plants obtain one insertion copy, and when the ratio is equal to 1, two insertion copies are obtained. (**B**) The relative expression of *TaXTH12.5a* in the roots of T3 homozygous transgenic Arabidopsis was determined by qRT-PCR. The relative level of transcripts was standardized by *Actin* (NM_001339262). The bar represents the average value of the triplicate. * Significantly different, *p* < 0.05, using Fisher’s minimum significant difference test.

**Figure 7 ijms-24-09886-f007:**
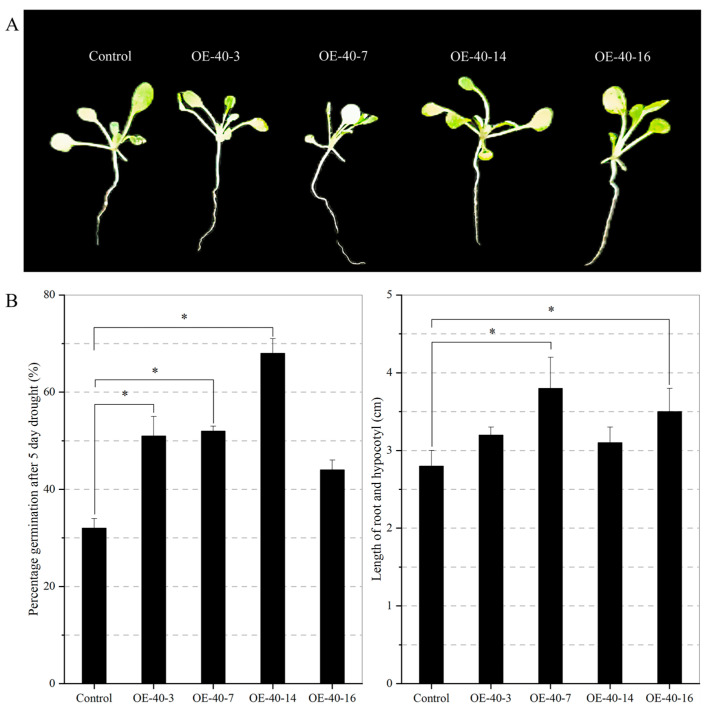
*TaXTH12.5a* transgenic plants showed higher germination rates and longer roots and hypocotyls under drought conditions. (**A**) The 3-day-old seedlings were dehydrated for 5 days. (**B**) The transgenesis and wild-type seeds were evenly spread on filter paper and subjected to a 5-day drought treatment under the same growth conditions. There were three repeated experiments and we counted their germination rates. (*n* ≥ 40). Root and hypocotyl lengths of the control and transgenic seedlings under drought conditions. (*n* ≥ 10). “*” indicates the difference between the transgenic and wild-type plants (*p* < 0.05).

**Figure 8 ijms-24-09886-f008:**
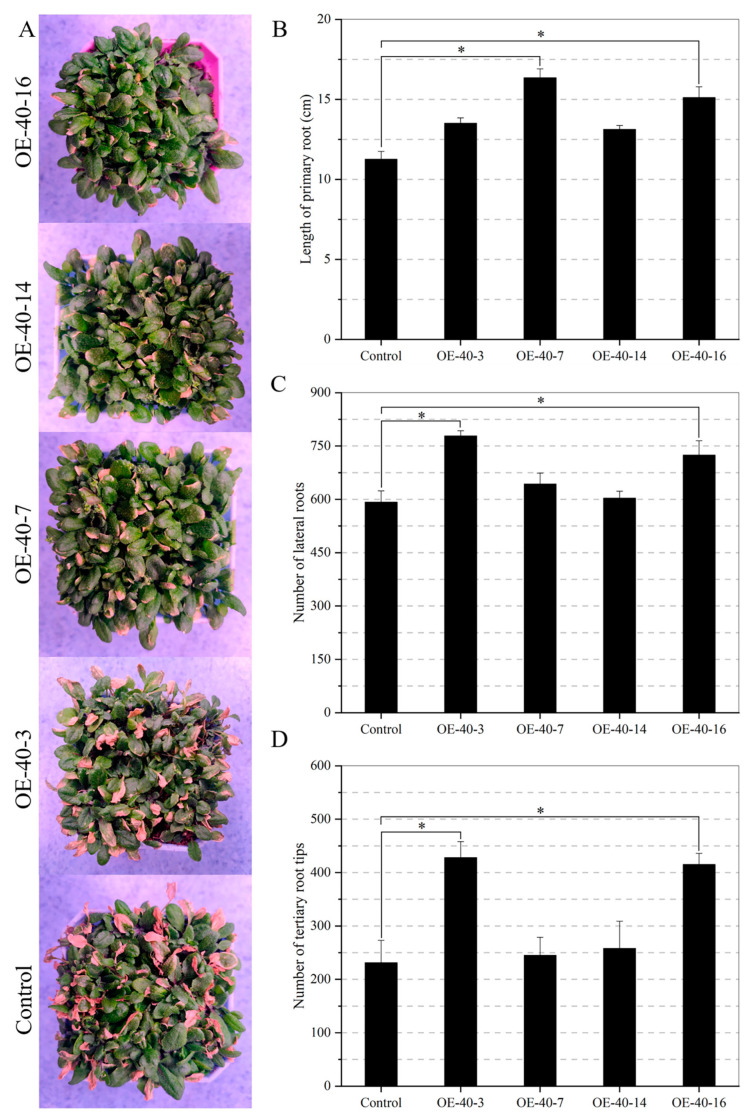
Arabidopsis *TaXTH12.5a* transgenic plants exhibit an enhanced drought tolerance phenotype by promoting root development. (**A**), effect of drought on Arabidopsis seedlings. Two-week-old seedlings were subjected to continuous drought for 5 days. (**B**), comparison of the primary root length of transgenic and control Arabidopsis under 5-day drought conditions. (**C**,**D**), effect of drought on the number of lateral roots and tertiary root tips of the transgenic plants. (*n* ≥ 40). * indicates the difference between the transgenic and wild-type Arabidopsis (*p* < 0.05).

## Data Availability

Not applicable.

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
