# Peer review of "A Surprising Diversity of Xyloglucan Endotransglucosylase/Hydrolase in Wheat: New in Sight to the Roles in Drought Tolerance"

_ijms, 2023, doi:10.3390/ijms24129886_

Round 1
Reviewer 1 Report
The authors identified all of the XET homologs from the sequence of the wheat genome.
NJ with Mega is a poor way to look at gene evolution. This program really compares percent similarity. Ought to do a parsimony or ML analysis. If you want to infer activity from these groups, it is important to do a thorough evolutionary analysis.
I don’t understand how the numbers correlate with the motifs (Section 2.3). Why are the numbers not listed consecutively? What do these numbers mean? Figure 3 is too small to understand anything. I got lost here.
The analysis of the transgenics was quite good. It’s nice to see multiple transgenics analyzed and the number of inserts evaluated.
What was the variation between germination rates (Fig. 7)? This is not indicated on the graphs, but differences were reported so multiple trials must have been done. A similar issue is present in Fig. 8.
Typographical errors are common, but the information the authors want to convey is clear. Here are the issues I found in the first 100 lines:
Line 21: spelled Arabidopsis
Line 53: remove f
Line 84: use conservation or homology
Line 100: Need a citation for Group 1 & II XTHs have conserved function
Author Response
Dear editors and reviewers:
Re: Manuscript ID: ijms-2391578 and Title: A Surprising Diversity of Xyloglucan Endotransglucosylase/Hydrolase in Wheat: New in Sight to the Roles in Drought Tolerance
Thank you for your precious comments and advice. Those comments are all valuable and very helpful for revising and improving our paper, as well as the important guiding significance to our researches. We have studied comments carefully and have made correction which we hope meet with approval. Revised portion are marked in red in the paper. We would love to thank you for allowing us to resubmit a revised copy of the manuscript and we highly appreciate your time and consideration.
Sincerely.
Junjie Han.
Q1: NJ with Mega is a poor way to look at gene evolution. This program really compares percent similarity. Ought to do a parsimony or ML analysis. If you want to infer activity from these groups, it is important to do a thorough evolutionary analysis.
Response: We appreciate your suggestions. We have revised our approach to the construction of the evolutionary tree in order to be more accurate and to meet the concerns of the reviewers. A phylogenetic tree was constructed using maximum likelihood (ML) by PhyML 3.0. The percentages of trees in which the associated taxa clustered together were determined from 500 bootstrap replications. We focused on identifying the TaXTH gene family and their potential role in improving plant drought tolerance. Therefore, we did not spend more energy on a more detailed evolutionary analysis.
Q2: I don’t understand how the numbers correlate with the motifs (Section 2.3). Why are the numbers not listed consecutively? What do these numbers mean? Figure 3 is too small to understand anything. I got lost here.
Response: We apologize for the inconvenience caused by the low image resolution. We will provide a higher-resolution image to meet the magazine's requirements. Here, I would be happy to explain the content of Figure 3A. We use MEME to analyze the conserved sequence (motif) of TaXTH proteins, extract the domain that reaches significance (P<0.05), and visualize it in TBtools. Different colors represent different motifs, and their values are sorted according to the P-value.
Q3: The analysis of the transgenics was quite good. It’s nice to see multiple transgenics analyzed and the number of inserts evaluated.
Response: We appreciate the reviewer’s positive evaluation of our work.
Q4: What was the variation between germination rates (Fig. 7)? This is not indicated on the graphs, but differences were reported so multiple trials must have been done. A similar issue is present in Fig. 8.
Response: We sincerely appreciate the reviewer’s suggestion. According to the reviewer’s comment, we provided more details to describe this mechanism as the following:
- We calculated the germination rate of transgenic or wild-type plants with three replicates (each with more than 40 plants) under drought conditions for five days. Germination enhancement of seeds by 5a and the results are shown in Fig. 7. We optimized the graph and marked the groups that achieved significant differences with '*'.
- As in the previous explanation, we counted the root indicators of at least 40 transgenic or wild-type plants in three replicates and conducted the statistical analysis. We optimized the graph and marked the groups that achieved significant differences with '*'.
Q5: Typographical errors are common, but the information the authors want to convey is clear.
Response: Thank you for your careful review. We are very sorry for the mistakes in this manuscript and inconvenience they caused in your reading. The manuscript has been thoroughly revised and rewritten by a native English speaker, so we hope it can meet the journal’s standard.
Furthermore, we have made additional modifications to the original manuscript in order to enhance its readability and better align it with the journal's requirements. We have already uploaded the revised manuscript and are eagerly awaiting approval from the reviewers. Once again, we would like to express our gratitude for the valuable comments provided by the reviewers. Thank you!
Reviewer 2 Report
The manuscript „A Surprising Diversity of Xyloglucan Endotransglucosylase/Hydrolase in Wheat: New in Sight to the Roles in Drought Tolerance” has been send to International Journal of Molecular Sciences to be considered for publication. The authors performed a silico analyses of XTH genes in wheat, performed a sequence analyses and analysed the expression of these genes under drought stress. The manuscript has a strong focus on the results, less on the concept and objectives of the study as well as the implications of the results and their interpretation.
Drought stress is for sure a severs issue in agronomy and to understand mechanisms to enhance the resilience of plants could be an important information to combat climate change. In this context the study is of high relevance, but the manuscript itself needs major improvements.
In general, English proof reading is recommended, the description and the objectives of the conducted studies needs to be improved. It is hard to follow the linkage between results, also because of their representation. It seems quite a lot of information and I am not sure, if all of this information is needed to provide a conclusion.
The discussion is strongly based on the results and repeats them in some aspect. A conclusion, consideration or the context for the drought stress response is not well described. A conclusive discussion would be helpful. Also the conclusions are very general.
Comments in more detail:
Material and methods:
- The identified TaXTH genes are putative XTHs – these would be considered in the manuscript
- Line 407: you mention, “they were renamed xxx” – does this implement, that XTH genes have already been described, maybe also functionally proved? If this is true, their names should be conserved. Please clarify this aspect.
- Did you somehow confirm the drought stress of plants?
- Please specify the used statistic
Results:
- Text in Figure 2 is very small – hard to read; also the resolution of Figure 3 could be improved
- PttXET16-34 – Please clarify – is not mentioned in Material and Methods
- Figure 4: expression analyses: In my opinion, it is not meaningful to cluster the heatmap between groups – you have a time-series of expression analyses with qPCR ? (I assume, not mentioned in the text or figure description). The heatmap should show results accordingly 0, 24, 48 and 72 h. In Material and methods, you mentioned the reference gene actin, but not your control group – these were control plants not drought treated at of the same age, or the expression at stage 0h? Additionally, the heatmap is based of fold changes or relative expression? These are just values, but there is no indication, if the differences are significant or not – you should somehow highlight significant values
- Line 202-204: it is not clear, what are the numbers referring to
- Figure 5: again, no statistic information, reference to which control samples, these are mean values? No standard deviation, scale of graphs different; the qPCR is based on which plants – seedlings or potted plants?
- More details in figure description could help – information is very limiting
- Figure 5: is it necessary to show all these genes? Maybe you could extract patterns – it is not clear, why this information is necessary
- Figure 6B: relative gene expression level as compared to the control? Add also standard deviations in bar charts
- Figure 7B this value is not well described – these are seeds under drought stress for 5 days and afterwards placed on media to germinate?
- Figure 8: it is not clear, how these data were obtained – condition of plant growth, replicates of pots, measurement of root length – please clarify in Materials and Methods
Minor comments:
Line 18: typo “ratess”
Line 21: typo “Arabidopsi and s”
Line 53: type “under f abiotic”
see above
Author Response
Dear editors and reviewers:
Re: Manuscript ID: ijms-2391578 and Title: A Surprising Diversity of Xyloglucan Endotransglucosylase/Hydrolase in Wheat: New in Sight to the Roles in Drought Tolerance
Thank you for your precious comments and advice. Those comments are all valuable and very helpful for revising and improving our paper, as well as the important guiding significance to our researches. We have studied comments carefully and have made correction which we hope meet with approval. Revised portion are marked in red in the paper. We would love to thank you for allowing us to resubmit a revised copy of the manuscript and we highly appreciate your time and consideration.
Sincerely.
Junjie Han.
Q1: The manuscript „A Surprising Diversity of Xyloglucan Endotransglucosylase/Hydrolase in Wheat: New in Sight to the Roles in Drought Tolerance” has been send to International Journal of Molecular Sciences to be considered for publication. The authors performed a silico analyses of XTH genes in wheat, performed a sequence analyses and analysed the expression of these genes under drought stress. The manuscript has a strong focus on the results, less on the concept and objectives of the study as well as the implications of the results and their interpretation.
Response: We are very grateful to the reviewers for carefully reading our paper and providing valuable feedback on any issues that may arise. We have sincerely reflected on these issues and are willing to try to improve them. The focus of this article is on the identification of the wheat XTH gene and its potential application in drought resistance. We conducted a comprehensive analysis of the expression levels of candidate XTH genes in seeds, roots, and stems under drought stress and validated the function of TaXTH through the performance of transgenic Arabidopsis under drought conditions. We have made further modifications and improvements in the manuscript to address the reviewers' concerns and gain their recognition and support. We are deeply grateful for the reviewer's suggestions and see them as a driving force for further research development. We sincerely hope these modifications make our research more accurate and convincing. Thank you again for the reviewer's review. We sincerely appreciate your valuable time and professional insights.
Q2: Drought stress is for sure a severs issue in agronomy and to understand mechanisms to enhance the resilience of plants could be an important information to combat climate change. In this context the study is of high relevance, but the manuscript itself needs major improvements.
Response: We sincerely appreciate the reviewers' recognition of our work. We apologize for any shortcomings in the article and will strive to improve it. Modifications have been made using ' Track Changes' to ensure transparency and traceability. We understand the importance of your feedback in enhancing the quality and credibility of the paper. We will actively incorporate your suggestions to improve the comprehensiveness and persuasiveness of the study.
Q3: In general, English proof reading is recommended, the description and the objectives of the conducted studies needs to be improved. It is hard to follow the linkage between results, also because of their representation. It seems quite a lot of information and I am not sure, if all of this information is needed to provide a conclusion.
Response: We apologize for the language issues present in the original manuscript. To address this, we sought assistance from a native English speaker with a relevant research background to improve the language presentation. For instance, specific improvements were made in lines 14-16, 193-194, and other manuscript sections.
Q4: The discussion is strongly based on the results and repeats them in some aspect. A conclusion, consideration or the context for the drought stress response is not well described. A conclusive discussion would be helpful. Also the conclusions are very general.
Response: Thank you sincerely for your valuable suggestion! We will make necessary modifications based on your concerns to ensure the research results are presented more accurately. The discussion section has undergone significant revisions to articulate our conclusions better. For example, we have significantly changed the overall coordination and discussion depth of paragraph "3.1. Characterization of TaXTH Gene Family." We hope these modifications meet your expectations and enhance the quality and credibility of the paper. We will sincerely consider every suggestion provided.
Q5: Line 407: you mention, “they were renamed xxx” – does this implement, that XTH genes have already been described, maybe also functionally proved? If this is true, their names should be conserved. Please clarify this aspect.
Response: We apologize for any potential misunderstanding caused by our previous statement. We can assure you that, apart from the already named TaXTH1-5 gene, no other gene has been assigned a name within the wheat XTH gene family. We carefully filtered out all hypothetical XTH genes in the wheat genome and assigned them numerical designations based on their chromosomal order. We will modify the sentence to reflect this: 'They were named xxx.' Once again, we sincerely appreciate your thorough reading and valuable feedback, which has proven immensely helpful.
Q6: Did you somehow confirm the drought stress of plants?
Response: Yes, before conducting this work, we evaluated the drought resistance of wheat and published relevant research in the journal (in Chinese). We subjected multiple wheat varieties to drought treatment under the same conditions and measured changes in multiple indicators, including rhizome length, the relative water content of leaves, photosynthetic characteristics, antioxidant enzyme activity, and osmoregulation substances. Through statistical analysis, we have confirmed the dynamic regulatory ability of plants under drought stress.
Q7: Please specify the used statistic.
Response: Thank you very much for the reasonable suggestions provided by the reviewers. We have followed the suggestion in line 532 of "4.9. Data analysis" and added statistical methods for missing data.
Q8: Text in Figure 2 is very small – hard to read; also the resolution of Figure 3 could be improved.
Response: We apologize for any inconvenience caused by the low-resolution images. We have acknowledged and rectified the related issues and included higher-resolution images in the manuscript to comply with the journal's requirements.
Q9: PttXET16-34 – Please clarify – is not mentioned in Material and Methods.
Response: We mentioned the relevant content in "Materials and Methods" on line 542 and submitted the relevant results in "Supplementary Files 1". We sincerely appreciate your understanding and patience.
Q10: Figure 4: expression analyses: In my opinion, it is not meaningful to cluster the heatmap between groups – you have a time-series of expression analyses with qPCR? (I assume, not mentioned in the text or figure description). The heatmap should show results accordingly 0, 24, 48 and 72 h. In Material and methods, you mentioned the reference gene actin, but not your control group – these were control plants not drought treated at of the same age, or the expression at stage 0h? Additionally, the heatmap is based of fold changes or relative expression? These are just values, but there is no indication, if the differences are significant or not – you should somehow highlight significant values.
Response: We sincerely thank the reviewers for their careful reading and excellent suggestions on our manuscript. These suggestions are beneficial for further improving our research. We have redesigned the heat map to better present gene changes at different time points (0, 24, 48, and 72 hours). Considering that we used many genes for qRT-PCR detection (71 genes), other graphics cannot clearly and aesthetically display the temporal trend of genes, so we chose heat maps as the display method. The values in the heat map are based on Log2 (relative expression). In addition, we supplemented the setting of the control group in the materials and methods section, which is a plant of the same age but not subjected to drought treatment.
Q11: Line 202-204: it is not clear, what are the numbers referring to
Response: We detected qRT-PCR on the expression patterns of 36 TaXTH genes in roots and shoots. The results showed that after 24 hours of drought stress, 21 genes showed high expression in the roots. We used both experimental and control samples 2−ΔΔCT values to determine the magnification relationships of expression levels.
Q12: Figure 5: again, no statistic information, reference to which control samples, these are mean values? No standard deviation, scale of graphs different; the qPCR is based on which plants – seedlings or potted plants?
Response: Thank you immensely for your corrections and invaluable suggestions, as they greatly aid us in enhancing the quality of our article. We wholeheartedly apologize for our oversight in not providing comprehensive information. In the meticulously revised manuscript, we have rectified the error of utilizing the average value to depict the changing trend in the original image. Additionally, we have provided a more elaborate exposition of the statistical information. We have also supplemented the article's description of the qPCR method, specifically emphasizing its implementation with potted plants. Once again, we express our gratitude for your corrections and your patience. Your feedback is highly valued, and we are committed to continuous improvement in our research endeavors.
Q13:More details in figure description could help – information is very limiting
Response: We have noticed the relevant issues and provided more detailed explanations for the captions in the manuscript.
Q14:Figure 5: is it necessary to show all these genes? Maybe you could extract patterns – it is not clear, why this information is necessary
Response: In order to further explore the function of TaXTH, we investigated the expression patterns of these XTH genes during drought stress. Through analyzing the qRT-PCR data of germinating seeds, we observed distinct expression profiles for each gene under different stress durations. We found varying durations of drought-induced differential expression of TaXTH genes. Specifically, TaXTH7.3b, TaXTH12.5d, and TaXTH9.2d were predominantly expressed at 48 hours of drought, while TaXTH7.1b, TaXTH9.1b, and TaXTH8.1b exhibited higher expression levels at 72 hours of drought. Consequently, we selected representative genes from these different time points, ultimately identifying TaXTH12.5a as a promising candidate for further research.
Q15: Figure 7B this value is not well described – these are seeds under drought stress for 5 days and afterwards placed on media to germinate?
Response: We express our sincere gratitude for the feedback received and acknowledge the value of the input provided. In response, we have made significant efforts to include more comprehensive explanations in the manuscript. After disinfection, the seeds were evenly spread in a box containing three layers of filter paper and then subjected to drought treatment following the procedure described in the "Materials and Methods" section. Set to repeat three times. We added D-sorbitol solution to the filter paper daily and removed any excess solution. After five consecutive days, we recorded the germination rate.
Q16: Figure 8: it is not clear, how these data were obtained – condition of plant growth, replicates of pots, measurement of root length – please clarify in Materials and Methods
Response: We apologize for not providing detailed information. We have supplemented the missing content in the materials and methods section. Thank you again for your understanding and support.
Furthermore, we have made additional modifications to the original manuscript in order to enhance its readability and better align it with the journal's requirements. We have already uploaded the revised manuscript and are eagerly awaiting approval from the reviewers. Once again, we would like to express our gratitude for the valuable comments provided by the reviewers. Thank you!
Reviewer 3 Report
A very interesting research/approach! I think that the present manuscript could be published prior to some minor revision, since I do not have any major concerns. It is a well-written manuscript and the authors are analytical and the tables that they are presenting indeed help the author very much. It is conscience and within the scope of the Journal.
I would recommend the authors to carefully read again the manuscript and correct some mistakes regarding the articles referenced. Some references do not match with what the authors report. Also, some linguistic/grammatical mistakes could be corrected .
some linguistic/grammatical mistakes could be corrected
Author Response
Dear editors and reviewers:
Re: Manuscript ID: ijms-2391578 and Title: A Surprising Diversity of Xyloglucan Endotransglucosylase/Hydrolase in Wheat: New in Sight to the Roles in Drought Tolerance
Thank you for your precious comments and advice. Those comments are all valuable and very helpful for revising and improving our paper, as well as the important guiding significance to our researches. We have studied comments carefully and have made correction which we hope meet with approval. Revised portion are marked in red in the paper. We would love to thank you for allowing us to resubmit a revised copy of the manuscript and we highly appreciate your time and consideration.
Sincerely.
Junjie Han.
Q1: A very interesting research/approach! I think that the present manuscript could be published prior to some minor revision, since I do not have any major concerns. It is a well-written manuscript and the authors are analytical and the tables that they are presenting indeed help the author very much. It is conscience and within the scope of the Journal.
Response: Thanks very much for taking your time to review this manuscript. We appreciate the reviewer’s positive evaluation of our work.
Q2: I would recommend the authors to carefully read again the manuscript and correct some mistakes regarding the articles referenced. Some references do not match with what the authors report.
Response: Thank you for your careful review. We carefully reviewed the manuscript and corrected the references. For example, the initial second reference was replaced, and more detailed information was tracked and displayed in "Track Changes" in the text.
Q3: Some linguistic/grammatical mistakes could be corrected.
Response: We apologize for the language problems in the original manuscript. The language presentation was improved with assistance from a native English speaker with an appropriate research background.
Furthermore, we have made additional modifications to the original manuscript in order to enhance its readability and better align it with the journal's requirements. We have already uploaded the revised manuscript and are eagerly awaiting approval from the reviewers. Once again, we would like to express our gratitude for the valuable comments provided by the reviewers. Thank you!
Round 2
Reviewer 1 Report
My comments have been addressed.
English usage is improved.
Reviewer 2 Report
The authors improved the manuscript and provided answers to the addressed comments of the first review. I highly appreciate their effort to modify the manuscript according to the recommendations. Especially the presentation of results improved including the applied statistics. The heatmap (figure 4) is still not conclusive for me, especially as all genes are suppressed at stage 0 - how come? at stage 0 control and treatment plants should be similar - at least that is what I usually expect in my experiments. Further more, it is not clear which genes are significantly modified and the results are presente "backwards" from 72 to 0h. The authors decided not to reduce the presentation of results in Figure 5 - I am still not sure, if all results needs to be presented, gene names are hard to read, so I miss the key information. Nevertheless, an explanation is provided in the answer to the review, but it is still a bit overwhelming and details are lost.
Improved.
Author Response
Dear editors and reviewers:
Re: Manuscript ID: ijms-2391578 and Title: A Surprising Diversity of Xyloglucan Endotransglucosylase/Hydrolase in Wheat: New in Sight to the Roles in Drought Tolerance
We would like to thank you for the opportunity to revise and resubmit our manuscript ijms-2391578, entitled " A Surprising Diversity of Xyloglucan Endotransglucosylase/Hydrolase in Wheat: New in Sight to the Roles in Drought Tolerance " by Han et al. We found the reviewers' comments to be helpful in revising the manuscript and have carefully considered and responded to each suggestion, corresponding changes to the resubmitted manuscript are highlighted in red. We would love to thank you for allowing us to resubmit a revised copy of the manuscript and we highly appreciate your time and consideration.
Sincerely.
Junjie Han.
Q1: The authors improved the manuscript and provided answers to the addressed comments of the first review. I highly appreciate their effort to modify the manuscript according to the recommendations. Especially the presentation of results improved including the applied statistics. The heatmap (figure 4) is still not conclusive for me, especially as all genes are suppressed at stage 0 - how come? at stage 0 control and treatment plants should be similar - at least that is what I usually expect in my experiments. Furthermore, it is not clear which genes are significantly modified and the results are present "backwards" from 72 to 0h.
Response: Thank you for the reviewers' attention and patience. Your suggestions are invaluable to our manuscript. After careful consideration, we have decided to adopt your initial suggestion. We have included the diverse expression patterns of 71 TaXTH genes in the manuscript and presented selected results in a horizontal bar chart, grouping genes with the same expression pattern. Supplementary diagrams provide additional detailed information. Initially, our goal was to concisely illustrate the relative expression changes of these genes using heat maps. However, we encountered challenges with parameter settings. As a result, we opted for a bar chart to represent these changes. We sincerely appreciate your suggestion, as it significantly enhances the readability of our article.
Q2: The authors decided not to reduce the presentation of results in Figure 5 - I am still not sure, if all results need to be presented, gene names are hard to read, so I miss the key information. Nevertheless, an explanation is provided in the answer to the review, but it is still a bit overwhelming and details are lost.
Response: Thank you sincerely for your valuable suggestion! We will make necessary modifications based on your concerns to ensure the research results are presented more accurately. We greatly appreciate the reviewer's suggestions and have decided to adopt your feedback. In the manuscript, we have chosen to display genes with similar expression patterns, while providing more detailed information in the supplementary document. We sincerely hope that these modifications will enhance the readability of our manuscript and earn the approval of the reviewers. Once again, we express our heartfelt gratitude for your patience and valuable advice.